Genome-wide discovered psychosis-risk gene ZNF804A impacts on white matter microstructure in health, schizophrenia and bipolar disorder

Mallas Emma-Jane 1 2
Carletti Francesco 3
Chaddock Christopher A. 1
Woolley James 4
Picchioni Marco M. 1 5
Shergill Sukhwinder S. 1
Kane Fergus 6
Allin Matthew P.G. 1
Barker Gareth J. 7
Prata Diana P. 7 8 diana.prata@kcl.ac.uk
1 Department of Psychosis Studies, Institute of Psychiatry, Psychology and Neuroscience, King’s College London, University of London , London , United Kingdom
2 Computational, Cognitive and Clinical Neuroimaging Laboratory, Division of Brain Sciences, Department of Medicine, Imperial College London , London , United Kingdom
3 Department of Neuroradiology, John Radcliffe Hospital, Oxford University Hospitals NHS Trust , Oxford , United Kingdom
4 Psychological Medicine, Royal Brompton & Harefield NHS Trust , London , United Kingdom
5 St Andrew’s Academic Department, St Andrew’s Healthcare , Northampton , United Kingdom
6 Department of Psychology, Institute of Psychiatry, Psychology and Neuroscience, King’s College London, University of London , London , United Kingdom
7 Department of Neuroimaging, Institute of Psychiatry, Psychology & Neuroscience, King’s College London, University of London , London , United Kingdom
8 Instituto de Medicina Molecular, Faculdade de Medicina, Universidade de Lisboa , Lisbon , Portugal
Fitzgerald Melinda
Electronic publication date: 2016 Feb 25
Publication date: 2016
Volume: 4
Electronic Location ID: e1570
Received 2015 Oct 17; Accepted 2015 Dec 15
Copyright: ©2016 Mallas et al.
Copyright year: 2016
Copyright holder: Mallas et al.
License: This is an open access article distributed under the terms of the Creative Commons Attribution License, which permits unrestricted use, distribution, reproduction and adaptation in any medium and for any purpose provided that it is properly attributed. For attribution, the original author(s), title, publication source (PeerJ) and either DOI or URL of the article must be cited.
License URL: https://creativecommons.org/licenses/by/4.0/

Keywords: Genome-wide association, White matter, ZNF804A, Psychosis, Fractional anisotropy, Diffusion tensor imaging, Schizophrenia, Bipolar disorder

Funding: UK National Institute for Health Research fellowship NIHR-PDF-2010-03-047 Fundação para a Ciência e Tecnologia Investigator IF/00787/2014 This research was supported by a UK National Institute for Health Research fellowship (NIHR-PDF-2010-03-047) and a Fundação para a Ciência e Tecnologia Investigator Grant (IF/00787/2014) to DP. The funders had no role in study design, data collection and analysis, decision to publish, or preparation of the manuscript.

==============================
Background. Schizophrenia (SZ) and bipolar disorder (BD) have both been associated with reduced microstructural white matter integrity using, as a proxy, fractional anisotropy (FA) detected using diffusion tensor imaging (DTI). Genetic susceptibility for both illnesses has also been positively correlated in recent genome-wide association studies with allele A (adenine) of single nucleotide polymorphism (SNP) rs1344706 of the ZNF804A gene. However, little is known about how the genomic linkage disequilibrium region tagged by this SNP impacts on the brain to increase risk for psychosis. This study aimed to assess the impact of this risk variant on FA in patients with SZ, in those with BD and in healthy controls.

Methods. 230 individuals were genotyped for the rs1344706 SNP and underwent DTI. We used tract-based spatial statistics (TBSS) followed by an analysis of variance, with threshold-free cluster enhancement (TFCE), to assess underlying effects of genotype, diagnosis and their interaction, on FA.

Results. As predicted, statistically significant reductions in FA across a widely distributed brain network (p < 0.05, TFCE-corrected) were positively associated both with a diagnosis of SZ or BD and with the double (homozygous) presence of the ZNF804A rs1344706 risk variant (A). The main effect of genotype was medium (d = 0.48 in a 44,054-voxel cluster) and the effect in the SZ group alone was large (d = 1.01 in a 51,260-voxel cluster), with no significant effects in BD or controls, in isolation. No areas under a significant diagnosis by genotype interaction were found.

Discussion. We provide the first evidence in a predominantly Caucasian clinical sample, of an association between ZNF804A rs1344706 A-homozygosity and reduced FA, both irrespective of diagnosis and particularly in SZ (in overlapping brain areas). This suggests that the previously observed involvement of this genomic region in psychosis susceptibility, and in impaired functional connectivity, may be conferred through it inducing abnormalities in white matter microstructure.

Introduction

Schizophrenia (SZ) and bipolar disorder (BD) are major psychiatric illnesses that have a profound effect on an individual’s mood, cognition and behavior. Lifetime prevalence of SZ and BD is about 4% (Bhugra, 2005) and 0.5% (Merikangas et al., 2007) respectively. Both illnesses are highly heritable: up to 80% (SZ) and 93% (BD), but their common and specific etiological and pathophysiological causes are poorly understood (Gurung & Prata, 2015).

One of the first genetic variants to achieve genome-wide significance for an association with both disorders, as well as independent replications, was the single nucleotide polymorphism (SNP) rs1344706 tagging an intronic region of the zinc-finger protein (ZNF) 804A gene (Gurung & Prata, 2015). The human ZNF804A gene, located on chromosome 2q32.1, codes for a protein consisting of 1210 amino acids. The protein contains one C2H2 type zinc-finger domain (Walters et al., 2010), which being typical of DNA/RNA-binding motifs, indicates that it may act as a transcription factor. Expressed in the brain (Bernstein et al., 2014), ZNF804A does seem to be involved in gene regulation (Donohoe, Morris & Corvin, 2010), including that of genes that are known to be SZ-candidate risk genes: COMT, DRD2, PRSS16 and PDE4 (Girgenti, LoTurco & Maher, 2012). It has been implicated in neurodevelopmental processes (Chung et al., 2010), cell adhesion, neurite outgrowth, dendritic branching and synapse formation (Hill et al., 2012), differentiation of oligodendrocytes and proliferation of oligodendrocyte progenitors (Riley et al., 2010).

The rs1344706 psychosis risk allele (i.e., A) of ZNF804A has lower binding affinity for proteins in the cell nucleus, such as transcription factors (Hill & Bray, 2011) and, potentially as a result of this, shows significantly increased expression compared to its counterpart (C allele) in healthy controls (Riley et al., 2010). Furthermore, this SNP appears to selectively modulate a novel mRNA isoform, ZNF804AE3E4 in the human fetal brain (risk allele homozygotes demonstrating lower expression than heterozygotes or non-risk homozygotes), with no effect on the full-length ZNF804A mRNA (Tao et al., 2014). The authors propose these findings suggest the ZNF804AE3E4 isoform may mediate the association of rs1344706 with psychosis. Nevertheless, the role of ZNF804A, or rs1344706, in psychiatric illness remains relatively unknown, with in vivo research of its involvement in brain structure and function highly warranted.

Neuroimaging studies of ZNF804A rs1344706 have not found an effect of the risk allele on regional brain activation, but rather on functional connectivity disruption between prefrontal regions (Walters et al., 2010; Esslinger et al., 2011; Walter et al., 2011; Paulus et al., 2013), which suggests its impact is on white matter (WM). Functional connectivity abnormalities are a common finding in BD and more so in SZ (Ohtani et al., 2014; Wang et al., 2014; Meyer-Lindenberg et al., 2005). WM abnormalities are also found in SZ (Makris et al., 2010) and BD (McDonald et al., 2005), including regional deficits common to both (McDonald et al., 2004; McIntosh et al., 2005; Kuswanto et al., 2012a). However, the impact of rs1344706 on WM volume, density and integrity is still unclear, as we reviewed elsewhere (Gurung & Prata, 2015). Fractional anisotropy (FA), measured using diffusion tensor imaging (DTI) is a putative proxy of WM microstructural integrity (Jones, Knosche & Turner, 2013). It is robustly found to be lower in SZ, and to a lesser extent, in BD, in a diverse range of brain regions (Ellison-Wright & Bullmore, 2009; Vederine et al., 2011). Reduced FA can be detected in very early stages of illness (Carletti et al., 2012), suggesting microstructural WM abnormalities are involved in the underlying neuropathophysiology of these diseases. FA, and other measures of WM microstructure (such as geodesic anisotropy and diffusivity), is reported to be highly heritable (Kochunov et al., 2015). Several studies also report FA abnormalities in first-degree relatives of patients with SZ and BD (Prasad et al., 2015; Skudlarski et al., 2013; Sprooten et al., 2013) with FA decreasing with increasing genetic liability to psychosis (Phillips et al., 2011; Emsell et al., 2013). This evidence provides support for FA being a potentially useful endophenotype for exploration of the mechanism of action through which ZNF804A rs1344706 is exerting increased disease risk.

The effect of rs1344706 on FA is still unclear, with three negative (Fernandes et al., 2014; Sprooten et al., 2012; Wei et al., 2013) and the following two positive association reports (Kuswanto et al., 2012b; Ikuta et al., 2014). Within the Chinese SZ population, risk allele homozygotes were found to have reduced FA in bilateral parietal lobes and left cingulate gyrus compared to non-risk allele carriers (Kuswanto et al., 2012b). Furthermore, within risk allele homozygotes, SZ patients showed decreased FA in the aforementioned areas, as well as the right medial temporal lobe (Kuswanto et al., 2012b). Consistently, in the healthy Caucasian population, reduced FA was associated with the risk allele A in a dose-dependent manner, in right parietal WM, left forceps minor and the anterior body/genu of the corpus callosum (Ikuta et al., 2014).

Taken together, the associations of reduced FA with SZ, BD, and the rs1344706 risk allele A, suggest that WM microstructural abnormalities may be part of the pathophysiological mechanism through which ZNF804A rs1344706 (or other polymorphism(s) in high linkage disequilibrium with it) increases risk for SZ ad BD. However, given that assessments of the impact of ZNF804A rs1344706 on WM microstructure have thus far yielded mixed results and are hard to compare given that they were found in different ethnicities or diagnosis statuses (Gurung & Prata, 2015), the present further study of the effect of rs1344706 on FA in a predominantly Caucasian and healthy as well as clinical sample, is highly warranted.

In the present study, we aimed to test two main hypotheses: (1) We aimed to assess the effect of ZNF804A rs1344706 genotype on FA in a predominantly Caucasian sample. We hypothesized that risk allele homozygotes (AA) would show reduced FA compared to C (cytokine) carriers, across diagnoses, at least in some WM regions; (2) We aimed to explore whether this genotype impacted FA differentially between the different diagnostic groups. Given that both allele A and reduced FA are correlated to SZ and, somewhat less strongly, to BD (Riley et al., 2010; Vederine et al., 2011; Skudlarski et al., 2013; Nortje et al., 2013; Schwab et al., 2013), we hypothesized that the genotype effect would be stronger in SZ and BD, compared to controls, and perhaps more so in SZ compared to BD. A whole brain approach, without a priori region-specific hypotheses, was taken given previous reports implicating a wide range of spatially extensive brain regions. In addition, we report the impact of SZ or BD on FA for completeness.

Methods

Participants

Our sample (n = 230) consisted of patients with SZ (n = 63), BD (type 1 or type 2; 77% of which with psychosis; n = 43) and controls (n = 124), which had participated in seven previous research studies (Allin et al., 2011; Chaddock et al., 2009; Chaddock, 2009; Kane, 2008; Kyriakopoulos et al., 2009; Picchioni et al., 2006; Shergill et al., 2007) at the Institute of Psychiatry, Psychology and Neuroscience (IoPPN), King’s College London. Individuals were collated from those sub-samples, with any relatives excluded. In the case of concordant monozygotic twins, one twin from each pair was removed at random; for discordant or dizygotic twin pairs, priority of inclusion was given to the individual with the genotype or, in this order of preference, the diagnosis, that was less frequent—in order to balance genotype and diagnostic group sizes as much as possible. Each participant was assigned to two groups: a diagnosis group (SZ, BD or control) and, after genotyping (see below), a genotype group (ZNF+ which included risk allele (A) homozygotes, or ZNF− which included heterozygotes and non-risk allele (C) homozygotes). Again, the merge within ZNF− had the purpose of maximizing counterbalance for this SNP (as is commonly practiced in the literature e.g., Kuswanto et al., 2012b; Schultz et al., 2014; Donohoe et al., 2011; Saville et al., 2015), given the very low frequency of allele C in the Caucasian population.

Table 1 Participant’s demographics per diagnosis and genotype groups.

Participants’ demographics (n = 230)		Diagnosis	ZNF804A rs1344706 Genotype	
		SZ (n = 63)	BD (n = 43)	Controls (n = 124)	Statistic, df, p-value	ZNF+ (AA; n = 105)	ZNF− (AC& CC; n = 125)	Statistic, df, p-value	
Age (SD)	33.78 (10.70)	41.07 (12.33)	35.79 (13.40)	F = 4.5, df = 2, p = 0.01a	36.94 (13.66)	35.62 (11.87)	t = − 0.77, df = 207.6, p = 0.44	
IQ z-scores (SD)b	−0.75 (2.89)	−0.87 (0.97)	−0.68 (3.51)	F = 0.70, df = 2, p = 0.50	−0.85 (3.35)	−0.33 (2.61)	t = 1.22, df = 197, p = 0.23	
CPZ- equivalent antipsychotics dose (SD)	696.94 (613.02)	341.60 (434.56)	n/a	t = 3.28, df = 104, p < 0.001a	641.93 (634.06)	484.45 (516.18)	t = − 1.41, df = 104, p = 0.16	
Years of education (SD)	13.74 (2.61)	14.81 (3.10)	14.90 (2.79)	F = 2.51, df = 2, p = 0.08	14.36 (2.73)	14.74 (2.95)	t = 0.85, df = 162, p = 0.40	
Sex (M/F)	50/13	18/25	67/57	χ2 = 17.24, df = 2, p = < 0.001	60/45	75/50	χ2 = 0.19, df = 1, p = 0.66	
Ethnicity (n)	Caucasian	46	40	104	χ2 = 13.90, df = 12, p = 0.31	79	111	χ2 = 20.86, df = 6, p = < 0.001 < 0.001	
	Black Caribbean	6	1	4	11	0	
	Black African	5	2	6	10	3	
	Central Asian	3	0	4	2	5	
	Mixed African-Caucasian	2	0	1	1	2	
	Eastern Asian	0	0	3	1	2	
	Other	1	0	2	1	2	
Handedness (n)	Right	62	38	112	χ2 = 5.79, df = 4, p = 0.22	93	119	χ2 = 3.88, df = 2, p = 0.14	
	Left	0	3	5	6	2	
	Mixed	1	2	7	6	4	
Genotype counts (%)	AA	27 (42.9)	19 (44.2)	59 (47.6)			
	AC	28 (44.4)	16 (37.2)	51 (41.1)					
	CC	8 (12.7)	8 (18.6)	14 (11.3)					
Notes.

a Statistically significant at p < 0.05.

b Scores of full scale IQ from the Wechsler Abbreviated Scale of Intelligence (WASI) (Wechsler, 1999), the Wechsler Adult Intelligence Scale–Revised (WAIS-R) (Wechsler, 1981) or the National Adult Reading Test (NART) (Nelson & Willison, 1991) were standardised to Z-scores to permit between-group IQ comparison. (The type of test used was balanced between diagnostic or genotype groups.)

n/a not applicable

ZNF+ High risk (AA genotypes)

ZNF− Low risk (AC& CC genotypes)

BD bipolar disorder

SZ schizophrenia

SD standard deviation

df degrees of freedom

The study was approved by the National Health Service South East London Research Ethics Committee, UK (Project “Genetics and Psychosis (GAP)” reference number 047/04). All subjects provided written informed consent at the time of participation. Patients were recruited from the South London and Maudsley National Health Service Trust (SLaM). Diagnosis, according to the criteria of the Diagnostic and Statistical Manual of Mental Disorders (DSM) 4th edition (American Psychiatric Association, 1994) was ascertained by an experienced psychiatrist using a structured diagnostic interview (with instruments detailed elsewhere, Prata et al., 2009). All SZ and BD patients were in a stable clinical state and all SZ and some BD were treated with antipsychotic medication (from which Chlorpromazine-equivalence was calculated, see Table 1). Exclusion criteria applied to all participants were a history of significant head injury and current (last 12 months) substance dependency according to DSM-IV diagnostic criteria. Controls were excluded if they had any personal or family history of a psychotic spectrum disorder. In order to follow the gold standard of experimental design that a control group must be matched to the experimental group on all variables except the one isolated for study, and avoid a biased ‘super-normal’ control group (Kendler, 2003), healthy participants with a previous diagnosis of any other Axis I disorder (or family history) were not excluded given these are frequently present in SZ and BD. Nevertheless, none were psychiatrically unwell or on any psychiatric medication at the time of participation.

Genotyping

DNA was extracted from blood samples or buccal swabs following a standard protocol (Freeman et al., 2003). The TaqMan SNP Genotyping Assay (Applied Biosystems, 2010) was performed for SNP rs1344706 (A/C) blind to any phenotype, at the Social Genetic and Developmental Psychiatry Centre (SGDP) lab, King’s College London. Possible genotype outcomes were thus C homozygous (CC, cytokine–cytokine), heterozygous (AC, adenine-cytokine) or A homozygous (AA, adenine–adenine). Distribution of Caucasian genotype frequencies (0.13 CC, 0.41 CA, 0.46 AA) was consistent with Hardy-Weinberg Equilibrium, calculated using Michael H. Court’s online calculator (Court, 2005) in Caucasian patients and controls (patients χ2 = 0.62, df = 1, p = 0.43; controls χ2 = 0.29, df = 1, p = 0.59) and African–American and Black Caribbean (patients χ2 = 0.29, df = 1, p = 0.77; controls χ2 = 0.03, df = 1, p = 0.87). Genotype counts are in Table 1.

Image acquisition

Magnetic Resonance Imaging (MRI) data were acquired using a 1.5T GE Signal LX system (General Electric, Milwaukee, WI, USA) in the Mapother House MR unit at the Maudsley Hospital, SLaM, London, UK, with actively shielded magnetic field gradients (maximum amplitude 40 mT/m1). A standard quadrature birdcage head coil was used for both radiofrequency (RF) transmission and signal reception. DTI data was acquired using a multi-slice peripherally-gated echo planar imaging (EPI) sequence, optimized for precise measurement of the diffusion tensor in parenchyma, from 60 contiguous near-axial slice locations for whole brain coverage, with isotropic (2.5 × 2.5 × 2.5 mm) resolution. At each slice location, 7 images were acquired with no diffusion gradients applied (b = 0), together with 64 diffusion-weighted images in which gradient directions were uniformly distributed in space. Acquisition parameters were: echo time (TE) = 107 ms, effective repetition time = 15 R-R intervals, duration of the diffusion encoding gradients =17.3 ms, with a maximum diffusion weighting = 1,300 s/mm2. Further details are given elsewhere (Jones et al., 2002).

DTI data processing

The raw DTI data were corrected for head movement and eddy current induced distortions, and brain-extracted using the Brain Extraction Tool (BET) (Smith, 2002) to exclude non-brain voxels. After visual inspection, the BET threshold was adjusted to 0.2 to ensure a balance between complete scalp removal and inappropriate erosion of brain tissue, not achieved with the default parameter of 0.5. FA images were created (with a mask defined by a binarised version of this brain-extracted image) by fitting a tensor model to the raw diffusion data using the Functional MRI of the Brain lab (FMRIB)’s Diffusion Toolbox (FDT) within FMRIB software library (FSL) as described elsewhere (Behrens et al., 2003).

Voxel-wise statistical analysis of the FA data was carried out using tract-based spatial statistics (TBSS) (Smith et al., 2006), part of FSL (Smith et al., 2004). All subjects’ FA data were aligned to FMRIB58_FA 1 × 1 × 1 mm standard space (an average of the FA images of 58 healthy adults) using the nonlinear registration tool FNIRT (Andersson, Jenkinson & Smith, 2007a; Andersson, Jenkinson & Smith, 2007b), which uses a b-spline representation of the registration warp field (Rueckert et al., 1999). The entire aligned dataset was then affine-transformed into a 1 × 1 × 1 mm MNI152 space, resulting in a standard space version of each subject’s FA image, from which the mean FA image was created and thinned, creating a mean FA skeleton. Each subject’s aligned FA data were projected onto this skeleton and the resulting data fed into voxel-wise cross-subject statistics.

Statistical analyses

Demographic differences between diagnostic or genotype groups were analyzed in Statistical Package for Social Sciences (SPSS, 2012) using independent t-tests, chi-square and analysis of variance (ANOVA). Scores of full scale IQ from the Wechsler Abbreviated Scale of Intelligence (WASI) (Wechsler, 1999), the Wechsler Adult Intelligence Scale–Revised (WAIS-R) (Wechsler, 1981) or the National Adult Reading Test (NART) (Nelson & Willison, 1991), were standardised to z-scores to permit between-group demographic comparison. The type of test used was balanced between diagnostic or genotype groups (Table 1).

The FSL Randomise tool (Anderson & Robinson, 2001) was used to perform permutation-based non-parametric inference on the skeletonized FA data at a threshold of 0.2 (TBSS default) with 10,000 permutations. The significance level was set at p < 0.05 after multiple comparisons correction using threshold-free cluster enhancement (TFCE) (Smith & Nichols, 2009), an approach that allows the significance of a target voxel to take into account not only the amplitude of the signal (in this case FA) but also the contribution of both the spatial extent and the magnitude of supporting voxels. To assess the main effect of genotype, of diagnostic group and their interaction on FA, an ANOVA-style design matrix was built with genotype (ZNF+ vs. ZNF−) and diagnosis (SZ, BD and controls) as the two independent variables. Mean FA in the largest cluster of each effect was graphically plotted for a visual overview. Cohen’s d measure of effect was calculated using mean FA of the largest cluster, to provide an approximate representation of the magnitude of effect found via TFCE analysis.

WM labelling, in accordance with JHU ICBM-DTI-81 WM Atlas (Mori et al., 2008), provided in FSL, was used to determine the anatomical location of significant FA clusters; only those with >1% probability were included in the cluster table. Where results were retrieved as ‘Unclassified’, labelling was carried out manually using the MRI Atlas of Human WM (Mori et al., 2005). Results were overlaid on MNI152 (1 mm) standard template and displayed in radiological convention.

Results

Demographics

Table 1 displays the participants’ demographics. BD patients (mean age = 41.1, SD = 12.3) were significantly (p < 0.05) older than SZ patients (mean age = 33.8, SD = 10.7; t(104) = − 3.2, p < 0.001 and controls (mean age = 35.8, SD = 13.4; t(165) = − 2.3, p = 0.02). There was no significant difference in age between controls and SZ (t(185) = − 1.11, p = 0.27). SZ patients (mean CPZ score = 696.9, SD = 613.0) had a significantly higher (t(104) = 3.3, p < 0.001) CPZ-equivalent score than BD (mean CPZ score = 341.6, SD = 434.6). There were significantly (χ2 = 17.2, p < 0.001) more males (50M:13F) in SZ than BD (18M:25F) or control (67M:57F) groups. There were no significant differences between diagnostic groups in IQ, years of education, ethnicity or handedness. Between ZNF+ and ZNF− groups, there were no significant differences in age, IQ, CPZ equivalents, years of education, sex or handedness. There was a lower proportion of Black African-American and Black Caribbean ethnicities in the ZNF− (n = 3) group compared to ZNF+ (n = 21) group (χ2 = 20.9, df = 6, p < 0.001), which was due to the A allele being naturally more common in these ethnicities than in the Caucasian population (Sherry et al., 2001).

Figure 1 Main effect of rs1344706 genotype on fractional anisotropy.

(A) FA was significantly lower in the high-risk (A homozygotes; ZNF+) group compared to the low-risk (C-carriers; ZNF−) group (p < 0.05, TFCE-corrected), irrespective of diagnosis in brain areas mapped in Fig. 2. Post-hoc analysis revealed that mean FA of ZNF+ was lower by half of a standard deviation (Cohen’s d = 0.47) than ZNF−, which equates to a ‘medium’-sized effect. (B) Within the largest cluster under a main effect of genotype cluster (44,054 voxels), the effect in SZ (Cohen’s d = 0.83) and BD (Cohen’s d = 0.89) was, all voxels averaged, ‘large’ while the effect in controls was ‘small’ (Cohen’s d = 0.2)—from a post-hoc analysis. As in subsequent figures, ‘Mean FA’ refers to the mean FA of the largest TFCE-corrected significant cluster, rather than to mean FA across the whole brain; with individual data points in “A” representing the mean FA of each individual within the same cluster.

Table 2 White matter tracts in clusters showing significant effects.

Cluster size (Voxels)	Z-statistic of cluster maximum	Cluster maximum (X, Y, Z coordinates)	White matter labelsa	
Main effect of ZNF804A rs1344706: ZNF + < ZNF−	
44,054	0.998	14	94	12	Genu of corpus callosum; Body of corpus callosum; R/L Anterior corona radiata; R Superior corona radiata; L Posterior thalamic radiation (include optic radiation); R/L External capsule; R/L Superior longitudinal fasciculus	
2,132	0.993	55	−40	−16	R Sagittal stratum (include inferior longitudinal fasciculus and inferior fronto-occipital fasciculus); R Superior longitudinal fasciculus	
1,214	0.993	34	−57	−45	Middle cerebellar peduncle; R Inferior cerebellar peduncle; R Superior cerebellar peduncle	
278	0.984	31	−47	−30	Middle cerebellar peduncle*	
218	0.98	45	−51	25	Unclassified	
216	0.979	10	32	51	Unclassified	
201	0.982	−8	39	−19	Genu of corpus callosum; L Anterior corona radiata	
182	0.986	9	−54	14	Unclassified	
109	0.968	−21	3	25	L Anterior limb of internal capsule; L Anterior corona radiata; L Superior corona radiata; L Superior fronto-occipital fasciculus (could be a part of anterior internal capsule)	
102	0.965	−16	15	−1	L Anterior limb of internal capsule	
90	0.974	34	−41	48	R Superior longitudinal fasciculus *	
78	0.973	7	14	37	R Cingulum (cingulate gyrus)	
69	0.964	−30	0	16	L Superior corona radiata; L External capsule	
63	0.982	−16	1	59	Unclassified	
63	0.971	−7	15	61	R Sagittal stratum (include inferior longitudinal fasciculus and inferior fronto-occipital fasciculus)*	
55	0.988	15	−3	61	R Corticopontine tract*	
36	0.966	−8	1	64	R Cingulum (hippocampus)*	
32	0.978	27	17	39	R Superior longitudinal fasciculus*	
28	0.976	35	19	−2	R Uncinate fasciculus *	
SZ-specific effect of ZNF804A rs1344706: SZ ZNF+ < SZ ZNF−	
51,260	1	14	−84	34	Genu of corpus callosum; Body of corpus callosum; Splenium of corpus callosum; R/L Anterior corona radiata; R Superior corona radiata; R Posterior thalamic radiation (include optic radiation); R External capsule; R Superior longitudinal fasciculus	
1,522	0.988	33	−57	−44	Middle cerebellar peduncle; R Superior cerebellar peduncle	
456	0.983	−8	−43	67	Unclassified	
261	0.989	−24	27	33	Unclassified	
117	0.976	−28	−6	−20	L External capsule; L Uncinate fasciculus	
110	0.994	34	−42	48	R Superior longitudinal fasciculus	
58	0.963	23	−12	−28	R Cingulum (hippocampus)	
53	0.963	−2	−36	−45	L Pontine crossing tract; Corticospinal tract; L Medial lemniscus	
49	0.975	3	−59	−12	R Uncinate fasciculus*	
36	0.964	−39	4	44	Unclassified	
34	0.979	16	−46	−24	R Inferior cerebellar peduncle	
29	0.967	11	27	20	R Cingulum (cingulate gyrus)	
29	0.983	−7	−51	−48	Unclassified	
22	0.961	41	34	6	R Sagittal stratum (include inferior longitudinal fasciculus and inferior fronto-occipital fasciculus)*	
21	0.976	−31	2	29	L Superior longitudinal fasciculus	
21	0.963	29	−4	−31	Unclassified	
Main effect of BD diagnosis: BD < Controls	
3,882	0.998	−17	25	23	Genu of corpus callosum; Body of corpus callosum; Splenium of corpus callosum; L Cerebral peduncle; R/L Retrolenticular part of internal capsule; R/L Anterior corona radiata; L Superior corona radiata; R/L Posterior thalamic radiation (include optic radiation); R Sagittal stratum (include inferior longitudinal fasciculus and inferior fronto-occipital fasciculus); L External capsule; L Superior longitudinal fasciculus	
Main effect of SZ diagnosis: SZ < Controls	
72,428	1	45	−10	−31	Genu of corpus callosum; Body of corpus callosum; Splenium of corpus callosum; R/L Anterior corona radiata; R/L Posterior thalamic radiation (include optic radiation); L External capsule; R/L Superior longitudinal fasciculus	
	
Notes.

a Only tracts with clusters at >1% probability, after threshold-free cluster enhancement (TFCE) correction, are included. White matter labels are provided in accordance with JHU ICBM-DTI-81 White Matter Atlas (Mori et al., 2008) using AtlasQuery in FSL unless marked with “*”, in which case they were based on MRI Atlas of Human White Matter (1st Edition by Mori et al., 2005—see methods) due to retrieval from AtlasQuery as ‘Unclassified’. When this was not possible, regions remained “Unclassified” as stated.

ZNF+ High risk (AA genotypes)

ZNF− Low risk (AC&CC genotypes)

BD bipolar disorder

SZ schizophrenia

FA fractional anisotropy (a putative proxy for white matter microstructural integrity)

Main effect of genotype on FA

Irrespective of diagnosis, the ZNF+ showed significantly reduced FA compared to the ZNF− group in the genu and body of the corpus callosum, bilaterally in the anterior corona radiata, external capsule, superior longitudinal fasciculus, posterior thalamic radiation, middle cerebellar peduncle and in the right inferior and superior cerebellar peduncle and left anterior limb of internal capsule, with the largest TFCE-corrected significant cluster encompassing 44,054 voxels (Fig. 1 and Table 2). A post-hoc analysis in SPSS showed that neither sex (F = 1.15, df = 1, p = 0.29) nor ethnicity (F = 0.58, df = 1, p = 0.45) explained FA variance in the largest cluster. Age was a significant contributor (F = 19.32, df = 1, p < 0.001) but when it was included in the model, genotype remained a significant explanatory variable (F = 12.27, df = 1, p < 0.001). There were no regions where FA was significantly lower in the ZNF− group compared to ZNF+ group.

For a better characterization of this main effect, a post-hoc inspection comparing the mean FA within the largest cluster, between genotype groups, in each diagnostic group, further revealed that this main effect was mainly driven by the genotype effect in SZ and in BD (Fig. 1B).

Effect of Genotype on FA in SZ

When we tested, across the brain, for an effect of genotype in each diagnostic group separately, we found no significant effect of genotype in controls or in BD (p < 0.05, TFCE-corrected). There was however a significant effect of genotype within the SZ group on its own in the genu, body and splenium of the corpus callosum, bilaterally in the anterior corona radiata, superior longitudinal fasciculus and uncinate fasciculus, right superior corona radiata, posterior thalamic radiation (including optic radiation), external capsule, superior cerebellar peduncle, inferior cerebellar peduncle, cingulum (cingulate gyrus) and the left corticospinal tract and medial lemniscus, with the largest TFCE-corrected significant cluster encompassing 51,260 voxels (Fig. 2 and Table 2). Again, taking the largest cluster as representative, neither sex (F = 0.50, df = 1, p = 0.49) nor ethnicity (F = 0.64, df = 1, p = 0.43) were significant predictors of mean FA, but age was so (F = 17.60, df = 1, p < 0.001). Nevertheless, as above, the effect of genotype on FA in this cluster remained significant (F = 5.80, df = 1, p = 0.02) after co-varying for age.

Figure 2 Effect of rs1344706 genotype on fractional anisotropy in schizophrenia.

(A) FA was significantly higher in ZNF+ group of SZ patients compared to the ZNF− group of SZ patients (p < 0.05, TFCE corrected) with a post-hoc large effect size given by a Cohen’s d of 1.01, i.e. a difference of one standard deviation between genotype groups, in the largest cluster (51,260 voxels). (B) Areas where FA was significantly lower in ZNF+ compared to ZNF− irrespective of diagnosis (i.e. main effect of genotype, plotted in Fig. 1B) are shown here in yellow. Areas where FA was significantly lower in ZNF+ compared to ZNF− in SZ alone, are shown in red. The overlapping areas where both these effects are significant are shown in orange.

Main effect of diagnosis on FA

SZ and BD showed, individually, significantly reduced FA compared to controls (p < 0.05, TFCE-corrected) across a spatially extensive cluster (Fig. 3), measuring respectively 72,428 and 3,882 voxels. The clusters overlapped extensively (Fig. 3 and Table 2) in the genu, body and splenium of the corpus callosum, anterior corona radiata (including the optic radiation) bilaterally, left external capsule and left superior longitudinal fasciculus. Neither ethnicity nor sex were significant contributors to the variance in the mean FA of the largest cluster of the ‘SZ < Control’ contrast (ethnicity: F = 0.73, df = 1, p = 0.39; sex: F = 2.79, df = 1, p = 0.10) or the ‘BD < Control’ contrast (ethnicity: F = 1.17, df = 1, p = 0.28; sex: F = 1.46, df = 1, p = 0.23) contrasts. Age contributed significantly to FA variance in both clusters, as expected given that it is well known to correlate with FA (Sullivan & Pfefferbaum, 2006), but the contribution of diagnosis remained highly significant as an explanatory factor of FA variance after controlling for age (for the ‘SZ < Control’ cluster: F = 26.99, df = 2, p < 0.001; for the ‘BD < Control’ cluster: F = 28.51, df = 2, p < 0.001). There was no significant difference in FA between patient groups, nor regions where FA was significantly decreased in controls compared to patients.

Figure 3 Main effect of diagnosis on fractional anisotropy.

(A) FA was significantly reduced in SZ compared to controls (marked **) and in BD compared to controls (marked *), p < 0.05, TFCE corrected. Post-hoc analyses in the largest significant clusters revealed a respective Cohen’s d of 0.91 and 1.19, both considered ‘large’. The difference in FA between SZ and BD was not statistically significant. Individual data points show mean FA value for each participant within the largest cluster of the effect. b. Areas in which FA was significantly lower in SZ compared to controls are shown in red and areas where FA was significantly lower in BD compared to controls are shown in blue. Each effect encompassed one spatially extensive cluster. The overlapping areas where both effects are significant are shown in purple.

Genotype x diagnosis interaction on FA

We found no WM areas where a genotype effect (in any direction) differed significantly between diagnosis groups (p < 0.05, TFCE-corrected), testing every possible diagnosis-wise comparison.

Discussion

We assessed the effect of ZNF804A rs1344706 genotype on FA, unprecedentedly, in a Caucasian clinical sample, as well as in health, and whether this genotype effect was different between diagnostic groups. For completeness, we also report FA differences between diagnostic groups. We found three statistically significant effects (p < 0.05, TFCE-corrected): (1) a main effect of genotype (irrespective of diagnosis), (2) an effect of genotype in the SZ patients group alone and 3) a main effect of diagnosis. We also detected no significant genotype by diagnosis interaction effects. Our results provide further support for the involvement of the GWA-discovered ZNF804A, in particular rs1344706 allele A, at least when in double-dose within a homozygous genotype, in inducing susceptibility to psychosis by demonstrating its effect in reducing FA in WM microstructure. We found unprecedented evidence in a predominantly Caucasian clinical sample, of an association between rs1344706 risk allele A and reduced FA in a wide WM network. Moreover, the opposite effect was found nowhere in the brain.

Our complementary post-hoc analyses using (for each individual) the mean FA across of the most significant TFCE-corrected clusters of each effect provide a representative measure of size magnitude and also allowed a better characterization of the significant main effect of genotype. Irrespective of diagnosis, the FA high-risk group (ZNF+, i.e., A homozygotes) was about half of a standard deviation lower (Cohen’s d = 0.48; Fig. 1A) than that of the low-risk group (ZNF−, i.e., C-carriers), which represents a ‘medium’-sized effect (Cohen, 1988). In the same ‘main effect of genotype’ cluster, both SZ and BD groups showed a ‘large’ effect of ZNF804A (SZ Cohen’s d of 0.83 and 0.89 respectively; Fig. 1B), which are effects almost as large as the diagnosis effects on FA (see below). In contrast, the effect of genotype in controls had a ‘small’ effect (Cohen’s d = 0.2). These effect sizes’ comparison serve to demonstrate that the effect of genotype in patients (both SZ and in BD) rather than in controls, was driving this main effect of ZNF804A rs1344706 on FA. A strong effect in patients is further supported, at least for SZ, by our findings of a large overlapping network (Fig. 2B) where an effect of genotype in SZ alone, is significant. Nevertheless, this difference in genotype effect size between diagnostic groups was not reflected in a significant TFCE-corrected genotype by diagnosis interaction in any area nor in the main genotype effect cluster.

The present main effect of genotype has been recently replicated in a Caucasian sample (Ikuta et al., 2014) who found that higher A allele dosage predicted reduced FA in right parietal WM and left forceps minor and, as in our study, the anterior body/genu of the corpus callosum. Importantly, both their and our independent findings in the (inter-hemispheric) corpus callosum provide the structural support to previous robust associations of this risk allele with reduced inter-hemispheric functional connectivity between dorsolateral prefrontal cortices during working memory, emotional face recognition and resting state (Esslinger et al., 2011; Esslinger et al., 2009). Indeed, the observation that a SZ risk allele could contribute to decreased prefrontal inter-hemispheric connectivity is consistent with the disconnection hypothesis of SZ, which has been particularly verified between the two hemispheres (Stephan, Baldeweg & Friston, 2006). Moreover, the risk allele has also been associated with increased fronto-temporal inter-hemispheric functional connectivity during working memory (Paulus et al., 2013; Esslinger et al., 2009), which was explained by this particular coupling being abnormally persistent during working memory in SZ (Meyer-Lindenberg et al., 2005). Furthermore, our observation that the genotype effect we found was at its highest in the genu and body of the corpus callosum is consistent with a previous report of inter-hemispheric connections being more heritable than intra-hemispheric or cortico-spinal ones (Shen et al., 2014). This evidence suggests that at least some of the genetic liability for psychosis may be acting on inter-hemispheric WM microstructure.

The allele-wise direction of the present genotype effect is not only consistent with neuroimaging and GWA findings, but also links particularly well with gene-transcription findings. The risk allele (A) has been associated with significantly higher gene expression than the C allele, in the human dorso-lateral prefrontal cortex of healthy controls, and, at trend level, in SZ (Riley et al., 2010). As alluded to above, this region has been implicated in abnormalities in function and connectivity associated with both SZ (Makris et al., 2005) and this polymorphism, and is directly reliant on a major WM tract where we report a large genotype effect: the superior longitudinal fasciculus. The same study (Riley et al., 2010) also found, bioinformatically, that the risk allele leads to the binding of two brain-expressed transcription factors (Myt1L and POU3F1/Oct-6), involved in oligodendrocyte differentiation and transition of pro-myelinating to myelinating Schwann cells. The C allele, however, results in binding of a non-brain associated transcription factor. Taken with the present and current findings, this suggests that the genomic region tagged by ZNF804A rs1344706 may be influencing risk for SZ and BD, or affecting symptom dimensions putatively more dependent on FA in SZ patients (see paragraph below), through differential provision of binding sites for transcription factors involved in WM tract myelination.

The same effect of ZNF408A rs1344706 was statistically significant in the isolated SZ group across widespread clusters which greatly overlapped with those where we found a main effect of genotype (irrespective of diagnosis), reaching a large effect size (Cohen’s d = 1.01; Fig. 3). No area showing a significant effect of ZNF804A was apparent for BD or controls in isolation. It is thus plausible that there is some other etiological factor(s) acting in SZ patients that increase(s) susceptibility to the effects of this risk variation on FA. Alternatively, rs1344706 is conferring risk to specific symptom dimensions in SZ that may be more dependent on WM microstructure in the reported areas. For example, healthy subjects have shown an association of the risk allele and higher Schizotypal Personality Questionnaire (SPQ) score elsewhere (Yasuda et al., 2011), with particular deficits in disorganization domains, although this has been challenged by an allele-wise incongruent finding (Stefanis et al., 2013). The fact that these genotype effects were larger than the effect of the same genotype on (the complex phenotype of) SZ or BD, typical of GWAs findings for mental illness (i.e., a ‘small’ odds ratio of 1.12) (Donohoe, Morris & Corvin, 2010) is expected given the rationale that intermediate phenotypes, or at least phenotypes less complex than behavior, are more closely related to genetic variation.

The present significant genotype effect in SZ patients is consistent with the uncorrected trend (Kuswanto et al., 2012b) found in Chinese SZ patients, in the parietal lobes bilaterally, the right temporal lobe and the left cingulate gyrus. However, the fact that the authors have not reported specific white fiber tracts impedes localized comparison with the present study. The authors also report an opposite trend in controls (to that in SZ) but it is of uncorrected statistical significance. In sum, our genotype-wise findings on FA are consistent with two studies that have found a positive association between rs1344706 and FA (Kuswanto et al., 2012b; Ikuta et al., 2014) and indirectly with nine studies that found an effect in functional connectivity (Esslinger et al., 2011; Walter et al., 2011; Paulus et al., 2013; Esslinger et al., 2009; Cousijn et al., 2015; Mohnke et al., 2014; Rasetti et al., 2011; Lencz et al., 2010; Linden et al., 2013), while three have failed to find an association (Fernandes et al., 2014; Sprooten et al., 2012; Wei et al., 2013).

Regarding main effects of diagnosis (controls vs. BD and SZ: Cohen’s d = 1.19 and 0.91, respectively), our findings replicate previous solid research showing that both BD (Vederine et al., 2011) and SZ (Ellison-Wright & Bullmore, 2009) are associated with reduced FA but with a larger difference in SZ (Skudlarski et al., 2013): although the effect sizes were similar, the FA reductions (TFCE-corrected) in SZ were almost 20 times more widespread than that in BD, compared to controls. Removing non-psychotic BD patients from the BD group does not alter this estimate much (Cohen’s d = 1.09). Putting diagnosis and genotype-wise effects in perspective, it should be noted that the (by far) largest significant clusters (p < 0.05, TFCE-corrected) both of the main effect of genotype and of the genotype effect in SZ were up to two thirds of that of the cluster size of the ‘SZ < Controls’ diagnosis effect (and more than 10 times larger than the ‘BD > Controls’ cluster; Table 2).

As a potential limitation, not all diagnostic groups were matched for age and sex. There is evidence of FA decreasing with age (Sullivan & Pfefferbaum, 2006) and perhaps differing by sex (see below). Nevertheless, if the effect of age would be confounding, BD patients would be expected to show decreased FA (as their age was higher) compared to SZ and controls, but they in fact show higher FA compared to SZ. Furthermore, age could not have confounded the finding of decreased FA in SZ compared to controls, which were well-matched age-wise, since co-varying for age in this situation would be expected to explain more of the error variance and thus further increase our power to detect a true group effect rather than decreasing it. Finally, although the SZ group contained a higher proportion of men than the control group, there is insufficient evidence to suspect that this would have artefactually created the well-replicated finding of decreased FA in SZ (Ellison-Wright & Bullmore, 2009; Reading et al., 2011; Scheel et al., 2013; Schneiderman et al., 2011) and BD (Vederine et al., 2011; Nortje et al., 2013; Lagopoulos et al., 2013). Although higher FA for men was found in the superior cerebellar peduncle, and for women in the corpus callosum (Kanaan et al., 2014), there is also evidence (Takao, Hayashi & Ohtomo, 2014) that after controlling for intracranial volume, sex differences seem to be due to differences in head size. Above all, these issues did not affect the main findings we report, i.e., the genotype effects, since the genotype groups were balanced for these demographic factors. Moreover, post-hoc analyses with the mean FA of the largest clusters of each significant contrast confirmed that the available demographic variables did not confound the effects of genotype or diagnosis.

Another limitation of FA studies is that, technically, reduced FA, although commonly taken as a proxy for reduced WM ‘integrity’ arising from deficient myelination, corresponds to heightened water diffusion within a voxel which, in rigor, can be attributed not only to reduced myelination but alternatively, or in conjunction, to several differences in WM microstructure: e.g., larger axonal diameter, lower axonal density, higher membrane permeability or lower intra-voxel orientational coherence of axonal fibers (Jones, Knosche & Turner, 2013). Thus, interpretation of FA should remain open. Nevertheless, in demyelinating diseases such as multiple sclerosis, the attribution of reduced FA to reduced myelination is immediate (Werring et al., 1999), and evidence has also been pointing to deficient myelination in SZ and BD (Du et al., 2013; Regenold et al., 2007), making the interpretation of FA reductions in SZ and BD as a proxy for WM microstructural integrity reductions increasingly plausible.

Conclusions

In conclusion, the present findings support previous evidence that homozygosis for risk allele A of SNP rs1344706 of ZNF804A confers risk for SZ and BD, and impaired functional connectivity (Esslinger et al., 2011; Walter et al., 2011; Paulus et al., 2013), by offering a possible pathophysiological mechanism whereby this genetic variant promotes reduced WM integrity in a widespread network. These results link particularly well with previous findings demonstrating that this risk variant, but not its counterpart, allows binding affinity for transcription factors that might disrupt myelination (Riley et al., 2010).

Supplemental Information

Supplemental Information 1 Raw Dataset

Click here for additional data file.

Additional Information and Declarations

Competing Interests

Author Contributions

Human Ethics

Data Availability

GJB received honoraria for teaching from General Electric Healthcare during the course of this research, and acts as a consultant for IXICO. No other author reports biomedical financial interests or declares potential conflicts of interest.

Emma-Jane Mallas performed the experiments, analyzed the data, wrote the paper, prepared figures and/or tables, reviewed drafts of the paper.

Francesco Carletti, Christopher A. Chaddock, James Woolley, Marco M. Picchioni, Sukhwinder S. Shergill, Fergus Kane and Matthew P.G. Allin contributed reagents/materials/analysis tools, reviewed drafts of the paper.

Gareth J. Barker reviewed drafts of the paper, provided guidance in DTI data processing and interpretation of findings.

Diana P. Prata conceived and designed the experiments, performed the experiments, contributed reagents/materials/analysis tools, wrote the paper, reviewed drafts of the paper.

The following information was supplied relating to ethical approvals (i.e., approving body and any reference numbers):

National Health Service South East London Research Ethics Committee, UK. Project “Genetics and Psychosis (GAP)”: ethical approval Ref No. 047/04.

The following information was supplied regarding data availability:

Figshare: figshare.com/s/8b73fe80737c11e5a8e806ec4bbcf141.

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
