# Peer review of "Genome-wide discovered psychosis-risk gene ZNF804A impacts on white matter microstructure in health, schizophrenia and bipolar disorder"

_PeerJ, doi:10.7717/peerj.1570_

## Round 0.1 · original submission · Major Revisions

The manuscript addresses an issue of interest however significant concerns have been raised by the Reviewers. In particular, both reviewers had concerns regarding the grouping of patient populations and the precise comparisons performed. The apparent lack of comparison and integration of findings relating genotype and disease state must be addressed in the revision, as well as the nature of the grouping of patient subsets for statistical analysis. This manuscript will require substantial revision and careful attention to both reviewer's comments before it can be deemed acceptable for publication in PeerJ
Please pay careful attention to the grammatical inconsistencies noted by the Reviewers as PeerJ does not offer editing services and improvement will be required before acceptance.

Reviewer 1 ·

Basic reporting

The authors prepared their manuscript carefully and the submission is adhere to the rules in "Basic Manuscript Organization "

Experimental design

some advice:
1. In line 47-50, the author wrote their data supports involvement of the corresponding genomic region in psychosis susceptibility and suggests that risk may be conferred through inducing abnormalities in white matter microstructure. But in this study, the author detected the FA value in three groups (control, SP and BD), The same effect of ZNF408 on FA exist in three groups. As for the significant difference of FA between control and patients can’t be accounted for by ZNF408A though the information of the difference of FA between ZNF+ and ZNF-group can be get to know. And this study can’t take light on susceptibility of SP and BD.The result of lower FA in ZNF+ group also can be find, but no AA distribute dominantly in SP or BD group, no effect on the susceptibility of SP and BD. The distribution of the genetype of rs1344706 in three groups should be depicted in the table 1.
2. In the present study, the author aimed to assess the effect of the genome-wide risk variant allele A of rs1344706 on tract-based regional FA,(line 122-123). But, the subjects were divided into two groups(AA and AC+CC)according to genetype. So the results can support AA genetype can decrease FA, not allele A.
3. “psychosis risk allele rs1344706 of ZNF804A appears to selectively modulate a novel mRNA isoform, ZNF804AE3E4 in human fetal brain (risk allele homozygotes demonstrate lower expression than heterozygotes or non-risk homozygotes) (line 79-81).How about the difference between heterozygotes or non-risk homozygotes? Considering question 3, maybe three groups should be divided according to genetype: AA, AC and CC.
4. “In order to remove the possibility of a ‘hyper-normal’ control group, healthy participants with a diagnosis of any other Axis I disorder were not excluded.” Any other Axis I disorder is OK? I think this is not enough rigorous.

Validity of the findings

some advice.
1. The patients with BD were in manic, depression or stable stage should be depicted in order to give readers more clear information.
2.In table 1, comparison of “Antipsychotics dose (in CPZ equivalent; SD)” in SP and BD,should be use two independent sample t-test ?
3. From line 47-50, “a large effect of ZNF804A in SZ (d=1.01)” was depicted, but no the effects in BD was depicted.

Additional comments

1. The author explored the difference of FA value among three groups(controls, patients with schizophrenia and patients with affective disorder. And analyzed association between ZNF804A polymorphism and FA value in two groups(homozygotes and heterozygotes or non-risk homozygotes ). It is important to explore the etiology either from imageology or from genetics. But two parts content weren’t combined logically. So, I adviced significant revision should be done. I think this study can’t answer the question put forward by the author themselves in this mauscript..
2. Please check “ZN4804A” in line 36 and 45.
3. Please check line 97. A redundant parenthese ?

Reviewer 2 ·

Basic reporting

In this manuscript the authors examine the role of rs1344706 in altering white matter integrity in the brain of control subjects relative to patients with schizophrenia and bipolar disorder.

This manuscript needs some improvement in order to be suitable for publication in this journal, and most of them relate to the clarity of communication.
1) Please rephrase the sentence at the end of paragraph 2 as it does not make sense. Also note that oligodendrocytes are post-mitotic cells and do not proliferate!
2) Commas are in unusual places in this manuscript and brackets often follow straight from the previous word, so some general editing is required.
3) In the introduction 'Reduced fractional anisotropy (FA; a putative(Jones et al., 2013) proxy of WM integrity) is another example of a sentence that is unclear. Also ZNF804A's is inappropriate scientific nomenclature.
4) In the introduction the authors state that of the two previous studies conducted on this topic one showed no change and the other a trend, so they almost talk the reader out of the need for this study upfront. They really need to better explain the rationale for their study on a primarily Caucasian population.
5) In the methods please clarify how the patients and controls were recruited instead of saying they were participants of a previous study - otherwise it is not possible to assess recruitment bias.
6) Please define terms such as DSM-IV, CC, CA, and AA. Also I would recommend the insertion of key sentences lacking field-specific jargon to make the paper more readable.

Experimental design

The submission is within the scope of the journal. As mentioned above the reason for the study, given previously unsuccessful findings, should be more clearly articulated. It is currently impossible to assess recruitment bias. Overall the imaging protocols appear valid. The study complies with ethical standards.

- In the first paragraph of the results please clarify what the means presented correspond to. i.e. mean of what? They state the distribution of African americans and black carribeans (written in full here!) within the ZNF+ and ZNF- groups. But no where do they clearly state the n values for each sub-group. It would be nice to know the comparisions were between ~20 BD patients that were ZNF+ and similar for the BD ZNF- group etc.

Validity of the findings

- Figure 1 is clear and the data looks compelling that FA is altered between controls and SZ and controls and BD.
- In the results the authors state 'irrespective of diagnosis, ZNF+ showed significantly reduced FA compared with ZNF- (Figure 2). However it is not clear whether they have statistically grouped all ZNF+ and ZNF- people, grouping the controls, SZ and BDs and just analyzing based on genotype, as the data is split on the graph, which implies they analyzed the effect of genotype within each diagnosis group. But then in Figure 3, they say that the genotype had an effect within the SZ group ... which implies that even though the data is split in Figure 2 the analysis was not. This is very misleading. Also the ZNF- SZ group in Fig 2 has a mean of 4.3, but that does not look the same in Fig 3.

The validity of the findings are difficult to assess due to the way in which the data are presented in the manuscript. It needs a re-work!

If reorganized and presented more clearly, the data may well be interesting.

---

## Round 0.2 · accepted · Accept

Your revised manuscript addresses the reviewer's comments in detail and is now clear with regard to purpose and execution.